# Relative Sensitivity of Core-Needle Biopsy and Incisional Biopsy in the Diagnosis of Musculoskeletal Sarcomas

**DOI:** 10.3390/cancers13061393

**Published:** 2021-03-19

**Authors:** Alexander Klein, Theresa Fell, Christof Birkenmaier, Julian Fromm, Volkmar Jansson, Thomas Knösel, Hans Roland Dürr

**Affiliations:** 1Musculoskeletal Oncology, Department of Orthopaedics, Physical Medicine and Rehabilitation, University Hospital, LMU Munich, 81377 Munich, Germany; t.fell@campus.lmu.de (T.F.); christof.birkenmaier@med.uni-muenchen.de (C.B.); julian.fromm@med.uni-muenchen.de (J.F.); volkmar.jansson@med.uni-muenchen.de (V.J.); hans_roland.duerr@med.uni-muenchen.de (H.R.D.); 2Institute of Pathology, University Hospital, LMU Munich, 81377 Munich, Germany; thomas.knoesel@med.uni-muenchen.de

**Keywords:** sarcoma, incisional biopsy, core needle biopsy, sensitivity, bone, soft tissue

## Abstract

**Simple Summary:**

A precise diagnosis is key in the correct treatment of sarcomas. However, which kind of biopsy should be done: A minimal invasive core needle biopsy (CNB) or an incisional biopsy (IB), yielding more tissue but requiring surgery? We compared the results of both methods after resection of musculoskeletal sarcomas in respect to the accuracy of the diagnosis. In total, 417 patients with 472 biopsies and final sarcoma diagnoses were included. The rate of unequivocal sarcoma diagnoses was 84.9% with CNB vs. 87.6% with IB (*p* = 0.465). The rate of repeat biopsies was higher with CNB as compared to IB (*p* = 0.003). There was no difference in the determination of the sarcoma subtype or the grade of malignancy. Sarcoma subtype, bone vs. soft tissue, and the biopsy technique utilized did not influence the sensitivity. The single exception to this was with chondrosarcomas, where IB was significantly superior to CNB (*p* = 0.024). Based on our data, the minimal invasive technique can be used without disadvantages in the majority of patients.

**Abstract:**

Background: There is no evidence as to the diagnostic value of the two most frequently used methods of biopsies in sarcomas: Incisional or core needle biopsy. The aim of our study was to evaluate the diagnostic sensitivity of the incisional and the core needle biopsy techniques in the diagnosis of bone and soft tissue sarcomas. Methods: We included 417 patients with a definitive diagnosis of bone or soft tissue sarcoma in whom a total of 472 biopsies had been performed. We correlated the results of the biopsies with the result of the definitive histopathological examination of the resected tumor. Dignity, entity, and grading (whenever possible) of the tissue samples were evaluated. Results: A total of 258 biopsies (55%) were performed in order to diagnose a soft tissue tumor and 351 biopsies (74.4%) were core needle biopsies. The number of repeat core needle biopsies, necessitated because of inconclusive histopathological results, was significantly higher (50 vs. 5; *p* = 0.003). We observed no significant difference regarding dignity, entity, and grading between the 2 different types of biopsies. Only with regards to the determination of dignity and entity of chondroid tumors, incisional biopsy was superior with statistical significance (*p* = 0.024). Conclusions: This study represents the largest study on biopsies for bone and soft tissue sarcomas. Based only on our results, we are unable to favor one method of biopsy and found high accuracy with both methods. Considering the potential complications, the added oncological risks of incisional biopsies and the ready availability of core needle biopsies, the latter, in our assessment, represents a valid and favourable method for bone and soft tissue sarcomas.

## 1. Background

Sarcomas are comparatively rare bone or soft tissue tumors, representing about 3% of all malignancies in adults [1]. Because of their rarity, diagnosis is often delayed. However, an accurate and timely diagnosis is essential for the timely start of the appropriate therapy [2,3]. Dependent on the location of the lesion, core needle biopsies (CNB) and incisional biopsies (IB) are the two main options for securing a diagnosis. Excisional biopsy should be only used in cases of small (<3 cm.) and epifascially located tumors [4].

While IB is considered the “gold standard” by many sarcoma experts, there is little evidence to support this standard [5]. An IB offers certain advantages, since a larger volume of tissue can be obtained and precise control of the incisional tract is possible, especially near vessels or nerves. The disadvantages of IB are the more frequent necessity for inpatient treatment with this procedure, higher cost, a higher risk of complications (e.g., hematoma), and a higher risk of potential contamination of the surrounding tissues [6]. In the case of CNB, ultrasound-, CT- or MRI-guidance is possible [7,8]. Already in 1991, Stoker et al. showed with 97% of primary correct diagnosis a very high sensitivity with CNB in the diagnosis of musculoskeletal lesions [9]. In addition, CNB can easily be performed on an outpatient basis. However, the diagnostic value of CNB is still being discussed controversially [10,11,12]. The decision for one of these two types of biopsy frequently depends on the infrastructure of a medical facility and/or the personal experience of its surgeons [6].

At our institution, CNB has traditionally played an important role in diagnosing musculoskeletal lesions. They have been and still are the primary method in our diagnostic and therapy algorithm.

The aim of this mono-centric retrospective study was to compare the sensitivity of CNB and IB in the diagnosis of soft tissue and bone sarcomas regarding a correct diagnosis of entity, dignity, and grading.

## 2. Methods

Inclusion criteria were:-Focusing on sarcomas the definitive diagnosis of a primary or locally recurrent soft tissue or bone sarcoma of the extremities, the pelvis, and the trunk after resection at our center. All benign and intermediate lesions had been excluded;-Biopsy performed at our musculoskeletal oncology center.

The key criterion for the inclusion of the patients in our cohort was the final diagnosis of a sarcoma. Our rationale for employing this kind of selection was that we intended to identify a homogenous cohort of sarcoma patients. Most of the published case series based their analysis on the complete patient collective, including suspected lesions [13,14,15]. The diagnostic algorithm of our Sarcoma Center requires a repeat biopsy in all suspicious lesions, whenever the first or the second tissue sample cannot confirm the diagnosis of a sarcoma. The interdisciplinary sarcoma board, including an experienced musculoskeletal radiologist, allows for the reassessment of the imaging studies and their correlation with histopathological findings. If the biopsy was negative, repeat imaging by means of MRI was repeated after an interval of 6–10 weeks. A new biopsy was then initiated in cases with a changing lesion. This algorithm ensures that the rate of false-negative diagnoses of sarcomas is reduced to a minimum.

Two experienced orthopedic oncologic surgeons performed all biopsies. We included 417 patients, treated between 2003 and 2017. These patients underwent a total of 472 biopsies. All patients received either magnetic resonance imaging or computed tomography. The patients with bone tumors (BT) received radiographs in addition. In our biopsy workflow, the feasibility of a CNB is generally assessed first. In cases with a close anatomical relationship of the tumor to vessels or nerves, we used CT- or ultrasound- guidance for obtaining a representative tissue sample. In cases with extended tumor necrosis or after failure of a CNB to provide a reliable diagnosis, we used an IB. In addition, in cases where some differential diagnoses were established beforehand and where more material was deemed necessary, a primary IB was performed.

After exact planning based on the cross-sectional imaging and palpation of the tumor, local anesthesia was applied. After performing a small stab incision of the skin, a core needle (14 G; 2.0 mm; MEDAX s.r.l. Unipersonale; San Possidonio, Italy) was used for soft tissue lesions. A Jamshidi needle (11 G; 3.1 mm; Fa. CareFusion LTD, San Diego, CA, USA) and fluoroscopic guidance was used in bone tumors. Then 2–3 tissue cylinders were sent for histopathological examination.

IB was performed under general anesthesia after identical planning. The skin incision was as small as reasonably possible with straight preparation to the lesion. For bone lesions, a guided 8–12-mm large core drill was used. Careful hemostasis was performed using a resorbable gelatin sponge (Fa. Aegis Lifesciences, Gujarat, India) to fill the bone defect. The majority of patients received a suction drain and an elasto-compressive bandage.

The term “entity” was defined as the type or group of musculoskeletal tumor according to the WHO classification. The term “dignity” refers to the differentiation between the benign and malignant tumors in the histopathological evaluation, also according to the WHO classification. The classification of “grading” was performed based on the classification of the FNCLCC (Fédération nationale des Centres de lutte contre le cancer, Paris, France); G1 corresponds to low-grade, G2 and 3 to high-grade, respectively). The grading of sarcoma was not feasible for every sarcoma subtype according to WHO classification. The classification of grading was not possible in sarcomas that had undergone neoadjuvant therapy. These cases were excluded from the sensitivity evaluation of grading.

In this retrospective study, the histopathology results obtained by biopsy and the final histopathological results after tumor resection were correlated. The histopathological evaluation was performed by 2 experienced pathologists. Every histopathological finding was discussed on the background of the imaging studies in the interdisciplinary board. In cases of inconclusive histopathological findings, an indication for repetition of CNB or an IB was discussed and performed accordingly.

The final histological findings were the basis of the database. The case of patient was graded as false-negative and non-sarcoma diagnosis for the statistical evaluation in case of inconclusive (benign or semimalignant) entity as result of histopathological examination. For statistical analysis, the data of all patients were included. Significance analyses were performed using the Mann–Whitney test, with a 95% confidence interval. The level of significance was set at less than 0.05. The data analysis software used was IBM^®^ SPSS^®^ Statistics 25.

## 3. Results

In total, 417 consecutive patients underwent 472 biopsies: 409 (86.7%) in primary tumors, 63 (13.3%) because of recurrent sarcomas. Of the patients, 224 (53.8%) were male, and 193 (46.2%) were female. The mean age was 52.3 years. Regarding the biopsies, 258 (55%) were performed in soft tissues, and 214 (45%) in bones. In total, 351 (74.4%) biopsies were CNB and 121 were IB (25.6%). Figure 1 shows the distribution of the sarcoma entities.

### 3.1. Failure Rate in Dependence of the Kind of Biopsy

In 352 of 417 patients (84.4%), a diagnosis of sarcoma was established with the first attempt (Figure 2). In total, 51 patients needed one repetition of biopsy, 2 patients repetitions. The percentage of repeat CNB, necessitated because of inconclusive results, was significantly higher (*n* = 50 of 351 (14.2%) vs. *n* = 5 of 121 (4.1%); *p* = 0.003) in comparison to repeat IB. In 404 (96.9%) cases, the biopsy finally showed a sarcoma. In 13 cases (3.1%), there were no signs of a malignant tumor in the histopathological examination of the tissue sample. In an interdisciplinary discussion based on clinical, radiological, and pathological findings, a malignant diagnosis was suspected. These cases underwent primary wide resection with the final diagnosis of a sarcoma.

### 3.2. Determination of Dignity

Comparing primary CNB and IB regarding their sensitivity with respect to the definitive malignant diagnosis, a rate of 83.3% (255/306 cases CNB) vs. 86.5% (96/111 cases IB) (*p* = 0.482) was found (Figure 2). In 53 cases with a non-malignant diagnosis or absence of tumor tissue from the first biopsy but with radiological characteristics of a malignant tumor, a repeat biopsy was performed. A second CNB was done in 45 cases (93.3% malignant results) and in 8 cases, an IB was performed as a second biopsy, 3 of which had undergone a primary CNB. All 8 showed malignant results with IB. Two patients with a second CNB required a third biopsy as an IB in order to arrive at a diagnosis. The analysis of false-negative biopsies showed no relevant specific factors, such as entity, location, or kind of tissue.

Included repeat biopsies, the total rate of correct CNB results was 84.9% (298/351 biopsies) vs. 87.6% with IB (106/121 biopsies; *p* = 0.465).

During the observational period, there were no patients with a malignant biopsy and a final diagnosis of a non-malignant tumor in the resection specimen.

### 3.3. Determination of Entity and Grading

Overall, in 472 biopsy samples, the entity determination was correct in 84.3% (102/121 biopsies) of the IB group compared to 80.1% (281/351 biopsies) in the CNB group (*p* = 0.304). A total of 187 of 472 biopsies (39.6%) were excluded from the grading evaluation because of neoadjuvant therapy. A correct grading, as well as the possibility, depending on the entity, was found in 53.4% of CNB (110/206 biopsies) vs. 65.8% of IB (52/79) (*p* = 0.058). An analysis of the different sarcoma subtypes also did not show any significant differences in the determination of dignity, entity, and grading (if feasible for the entity) between CNB and IB with a single exception (Table 1). This was the determination of dignity of chondroid tumors (enchondroma vs. chondrosarcoma) by means of CNB. IB had a significantly higher specificity in those cases (88.9% (18/20 cases) vs. 66.7% (38/57; *p* = 0.024).

### 3.4. Differences between Bone and Soft Tissue Sarcomas

The type of tissue did not influence a correct diagnosis of malignancy (83.6%; 179/214 biopsies) in bone sarcoma vs. 87.2%; 225/258 biopsies in soft tissue sarcoma; *p* = 0.272). 196 biopsies in soft tissue sarcomas (STS) were performed as CNB. In 171 (87.2%), a correct result regarding malignancy was made. In bone sarcomas 155 (72.4%) of 214 biopsies were performed as CNB (Table 2). In 127 (81.9%), a correct result regarding malignancy was established.

In soft tissue sarcomas, the rate of primarily correct histopathologic diagnoses was identical between CNB and IB (79% in both groups), and also dignity or grading were not different within this group between both types of biopsy. In bone sarcomas, we also did not observe significant differences between CNB and IB regarding a correct diagnosis of entity and dignity. The evaluation of grading was done after exclusion of osteosarcomas and Ewing sarcomas because of neoadjuvant therapies and was hence limited to chondrosarcomas with significantly higher sensitivity for IB (*p* = 0.024).

In cases of local recurrence, the biopsy was significantly more sensitive in comparison to primary diagnosis of sarcoma (95.2 vs. 84.1%; *p* = 0.019) with CNB and IB showing the same results.

Thus, in total, 65 (of 417 cases; 15.6%) of all primary biopsies returned false-negative results (i.e., benign or no tumor tissue) and in these cases, a second or even a third biopsy was necessary to establish the correct diagnosis. There were no significant differences between CNB and IB with regards to the determination of malignancy, entity, and grading of the sarcomas, with one exception: In cases of chondrosarcoma, IB was superior to CNB.

## 4. Discussion

It is essential to obtain an adequate amount of tumor tissue when performing a biopsy in order to establish the correct histopathological diagnosis. There is general consensus that, in this sense, a fine needle aspiration biopsy is not a reliable method in bone sarcomas [16]. In this context, IB has been considered the gold standard for decades [17]. However, good results of CNB in the diagnosis of sarcomas are described. A number of publications describe the accuracy of CNB in bone and soft tissue lesions. Two large series include several hundred cases: Yong et al. achieved a diagnostic accuracy (entity) of 89% in 509 cases of bone and soft tissue tumors [18]; Ng et al. 77.2% in 432 soft tissue tumors [13]. CNB is easily available and less invasive than IB.

### 4.1. Patient Selection

Other studies have compared CNB and IB in the diagnosis of bone tumors of different dignities [6,19]. Our study included 417 cases with 472 consecutive biopsies in sarcomas only. Patient inclusion into our study was based on the final sarcoma diagnosis. This kind of cohort selection is not commonly used. However, this strategy allows for the building of a homogeneous study group. The infrastructure of a Sarcoma Center with highly specialized radiologists, surgeons and pathologists and regular case reviews leads to a differentiated and detailed approach to findings with sarcoma-suspicious lesions and negative histopathological results [20,21]. This workflow reduces the risk of false-negative biopsy results in sarcoma patients to the practically achievable minimum.

### 4.2. Disadvantages of Core Needle Biopsy

The authors are well aware of the fact that due to the small sample sizes obtained with CNB, a major disadvantage of this strategy was a loss of vital tissue for research, i.e., storage in a tissue bank. CNB (14.2%) had a significantly higher failure rate, when compared to IB (4.1%) in our cohort. The most common reason for failure was a non-representative tissue sample from the periphery of the lesion. Other authors have described the difficulties with the technical implementation of CNB as a frequent cause of the failed biopsy [15]. Our standard instrument for CNB is a 14-gauge Tru-Cut needle, according to the recommended guidelines [4]. In addition, and in order to improve the accuracy of CNB, currently we perform these biopsies under ultra-sound guidance more frequently in selected cases (small tumor, non-palpable location). Andreou et al. reported the inferior results (higher rate of local recurrences: 4.2% vs. 10.1%; *p* = 0.001) in patients, who underwent biopsies outside experienced centers [22]. The repetition of a biopsy could adversely affect the outcomes of treated patients according to these results. However, this result is mainly based on IB with a higher risk of contamination. The argument of a faster diagnostic procedure by CNB is put into perspective in 14% of patients needing a second biopsy. However, repetition of the new biopsy is normally within one week possible. There is no evidence for the influence of symptom duration on the oncologic outcome: None of citied studies was able to show a negative effect of longer symptom duration on overall survival of sarcoma patients [23,24,25].

### 4.3. Results in Respect to Tissue Type and Entity

The comparison between bone and soft tissue sarcomas showed similar results in both groups (83.6% vs. 83.3%). Some studies suggested a worse sensitivity for malignancy in bone tumors with CNB as opposed to IB [26,27]. In another study, the diagnostic accuracy of CNB’s was 100% in bone tumors [14]. Our results as compared to other authors [11,15] were less convincing (15% rate of false-negatives). The only exception was the subgroup of chondrosarcoma patients. Initially, 18% of chondrosarcomas were incorrectly diagnosed as benign tumors: 33.3% in the CNB group and 11.1% in the IB group. In cartilaginous tumors, IB therefore seems to be the better choice, whereas in all other bone lesions, CNB and IB are equivalent [28]. Similar results have already been indicated by other authors [29,30]: Roitman et al. demonstrated an impressive failure rate of CNB with the grading of chondrosarcomas (64% in pelvic bones). This makes the imaging all the more important in assessing chondroid tumors [31].

The comparison of CNB and IB in the subgroup of STS shows a homogenous result. The accuracy rate of IB and CNB (76.4% vs. 73%) is comparable with the international literature [32,33] and does not show any significant differences. Some authors have reported difficulties with the diagnostic procedure in certain soft tissue sarcomas, like angiosarcoma and synovial sarcoma [19,34]. In total, 258 biopsies in STS were done in our cohort. We were unable to identify subgroups of STS, which had a higher probability of correct diagnosis by one particular kind of biopsy. The inclusion of local recurrences in this study has to be discussed. Knowing the primary tumor might facilitate the final diagnosis. The diagnosis of the entity in recurrent tumor might be easier. However, in some cases we observed changes from a more distinct lesion to an undifferentiated sarcoma. in addition, systemic therapy might induce a change in the tumor’s biology during the course of treatment, so that the secondary tumor can differ from the primary sarcoma. The biopsy of the suspected recurred tumor is recommended for these reasons [35].

### 4.4. Advantages of CNB Regarding Local Recurrence

The risk of local recurrence in dependence of the type of biopsy is also of significant importance. Barrientos–Ruiz at al. analyzed the oncological outcomes of 180 sarcoma patients with different kinds of biopsies and the contaminations of biopsy tracts. Their results were: The contamination of the biopsy tracts was significantly higher in the cases of IB (32% vs. 0.8%) [36] and these were associated with a higher number of local recurrences [37]. The lower risk of local recurrence after CNB was confirmed in other studies [38,39]. At many musculoskeletal oncology centers, CNB is therefore favored over IB [6,12,36].

### 4.5. Study Design and Quality

The retrospective non-randomized design of this study limits the power of our findings. As stated in the method section due to the preselection of cases with difficult differential diagnosis to IB a certain degree of selection bias has to be acknowledged. Despite these limitations, this study is the largest mono-centric series comparing CNB to IB in sarcoma patients. Due to only two surgeons performing all procedures, the techniques are comparable and the patient group is very homogenous. The recently published meta-analysis from Birgin et al. essentially confirms our results [5]: The evaluation of biopsies in 2680 patients with sarcomas (17 studies analyzing CNB and IB) ranges CNB superior to IB.

### 4.6. Summary

In summary, CNB is at least equivalent to IB. As the higher risk of complications in IB vs. CNB is well known [40], in consideration of a higher risk of complications and possibly worse oncological outcome with IB, CNB is a valid and favorable method of biopsy in the diagnostics of bone and soft tissue sarcomas. The only exception are cases of cartilaginous tumors, where IB should be preferred. A comparison of local recurrence and complications in both types of biopsies is necessary in future studies.

Based on these and previous findings, we have established an internal algorithm for the diagnostic workup in sarcoma cases: The primary biopsy method is CNB, when appropriately guided by sonography or computed tomography. Only in cases of cartilaginous tumors, should IB be preferred.

## 5. Conclusions

Based on our results, both CNB and IB have a high diagnostic accuracy in suspected sarcomas of the musculoskeletal system. Considering the potential complication and the oncological risks of IB and the better availability of CNB, the latter could be a more favorable method in the diagnosis of musculoskeletal sarcomas, even if in 15% of patients a second CNB was necessary.

## Figures and Tables

**Figure 1 cancers-13-01393-f001:**
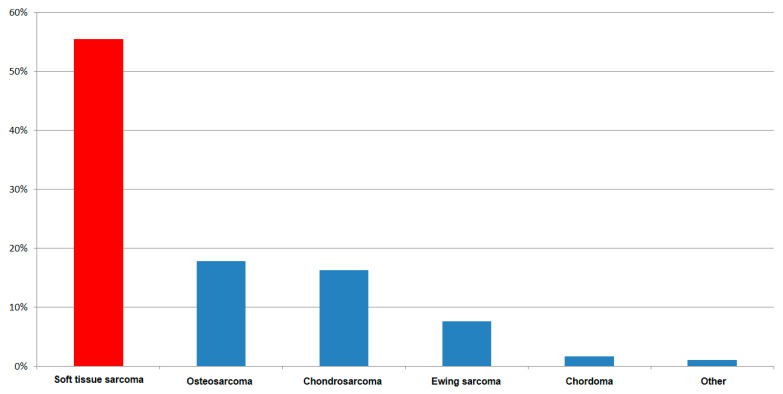
Tumor entities in 417 patients.

**Figure 2 cancers-13-01393-f002:**
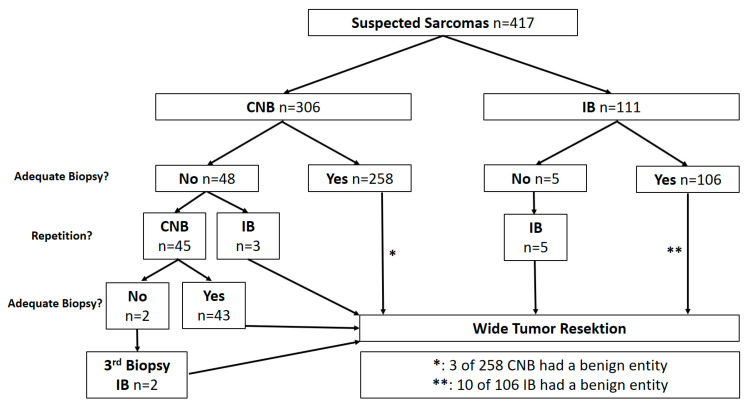
Characteristics of biopsies (CNB: Core needle biopsy, IB: Incisional biopsy).

**Table 1 cancers-13-01393-t001:** Sensitivity of biopsy kinds in different subtypes of sarcomas (CNB: Core needle biopsy; IB: Incisional biopsy; bolded *p*-value is a significant difference, *p* < 0.05).

Subtypes of Sarcomas	Dignity			Entity			Grading		
	CNB	IB	*p*	CNB	IB	*p*	CNB	IB	*p*
Osteosarcoma	39/52 (75%)	26/32 (81.3%)	0.506	45/52 (86.5%)	28/32 (87.5%)	0.899			
Chondrosarcoma	38/57 (66.7%)	18/20 (88.9%)	**0.024**	45/57 (78.9%)	18/20 (88.9%)	0.270	29/57 (50.8%)	14/20 (70.0%)	0.181
Ewing Sarcoma	26/32 (81.3%)	4/4 (100%)	0.343	26/32 (81.3%)	4/4 (100%)	0.343			
Myxofibrosarcoma	14/19 (73.7%)	5/6 (83.3%)	0.629	14/19 (73.7%)	5/6 (83.3%)	0.629	6/13 (46.2%)	2/4 (50.0%)	0.893
Liposarcoma	27/37 (73.0%)	17/21 (81.0%)	0.495	30/37 (81.1%)	17/21 (81.0%)	0.990	16/26 (61.5%)	12/19 (63.2%)	0.912
MPNST	13/15 (86.7%)	3/5 (60.0%)	0.197	13/15 (86.7%)	4/5 (80.0%)	0.718	2/7 (28.7%)	1/2 (50.0%)	0.571
Synovialsarcoma	6/6 (100%)	8/9 (88.9%)	0.398	6/6 (100%)	9/9 (100%)		2/2 (100%)	2/3 (100%)	
Leiomyo-/Rhabdomyosarcoma	21/23 (91.3%)	5/5 (100%)	0.494	21/23 (91.3%)	5/5 (100%)	0.494	6/10 (60%)	3/3 (100%)	0.188
Epithelioid Sarcoma	7/9 (77.8%)	2/2 (100%)		8/9 (88.9%)	2/2 (100%)	0.621	5/6 (88.3%)	0	

**Table 2 cancers-13-01393-t002:** Characteristics of biopsies and their result (STS: Soft tissue sarcoma; BS: Bone sarcoma; CNB: Core needle biopsy; IB: Incisional biopsy).

	Kind of Biopsy	*n*	*p*
Total Number of Biopsies		472	
Kind of Tissue	STS	258 (55%)	
	BS	214 (45%)	
Kind of Biopsy	CNB	351 (74.4%)	
	IB	121 (25.6%)	
Dignity Clarified	CNB	84.9%	
	IB	87.6%	0.465
Biopsy Repeated	CNB	50 (14.2%)	
	IB	5 (4.1%)	0.003
STS Confirmed Dignity	CNB	73%	
	IB	76.4%	0.976
STS Confirmed Entity	CNB	79.1%	
	IB	79%	0.824
STS Confirmed Grading	CNB	60%	
	IB	64.1%	0.610
BS Confirmed Dignity	CNB	81.9%	
	IB	88.1%	0.273
BS Confirmed Entity	CNB	76.8%	
	IB	84.7%	0.201

## Data Availability

The data presented in this study are available by corresponding author of this study.

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
