# Peer review of "Relative Sensitivity of Core-Needle Biopsy and Incisional Biopsy in the Diagnosis of Musculoskeletal Sarcomas"

_cancers, 2021, doi:10.3390/cancers13061393_

Round 1
Reviewer 1 Report
The authors present the results of a retrospective analysis comparing the diagnostic sensitivity of incisional (IB) and core-needle biopsy (CNB) in patients with bone and soft tissue sarcoma. While the topic is interesting and has been controversially discussed in the sarcoma community for many years, the manuscript has several weaknesses, which the authors need to address in major revisions, before it can be accepted for publication. The most important of these weaknesses concerns the fact, that the discussion and the conclusion of the authors convey a somewhat biased impression - for such a controversial topic, a more balanced discussion presenting both sides of the arguments for and against CNB or IB are necessary. The authors did not perform a prospective, randomized study, but a retrospective analysis. As a result of this study design, they can actually only conclude that their algorithm for choosing to perform CNB and IB appears to be adequate for most aspects of diagnostic accuracy, although patients undergoing CNB had a significantly more repeat biopsies, something that the authors should emphasize more in their conclusions. What they cannot do is draw conclusions regarding the "diagnostic method of choice" in patients with bone and soft tissue sarcomas in general. Furthermore, the authors base their conclusions on the hypothetical higher rate of complications and risks of IB compared to CNB, however they have performed no respective analyses in their cohort, which seems to be a missed opportunity.
Further issues that need to be addressed (both in the abstract and in the main manuscript) are:
- Introduction: the authors should provide appropriate references for their statements (e.g. lines 34-37, 40-44, 49-50 etc.).
- Introduction (and discussion): When the authors present the results of previous studies, they tend to be very vague (e.g. "Stoker et al showed a very high sensitivity with core-needle biopsies in the diagnosis of musculoskeletal lesions"). I would recommend that the authors go through their introduction and discussion and provide more details when presenting previous studies (e.g. what is "very high" sensitivity and in what setting with how many patients did Stoker et al show this?).
- Introduction: The authors mention that: "The Department of Orthopaedic Oncology is the longstanding constituent of a dedicated Sarcoma Center. The interdisciplinary approach for the discussion of all newly diagnosed sarcoma cases as well as of all cases with suspicious lesions but without positive histological findings on biopsy has proven very valuable over the years.". This statements are unnecessary and not supported by facts or literature - they could therefore be interpreted by some readers as advertisement. Please remove them.
- Introduction, lines 61-66 - definitions of terms should be included in the Methods section, not the introduction section of the manuscript. Furthermore - the authors mention benign and malignant tumors. In terms of dignity, how did they classify locally advanced/rarely metastasizing bone and soft tissue tumors, which are considered neither benign nor malignant?
- Methods, lines 68-71: the authors appear to have only included patients who had the "definitive diagnosis of sarcoma" after resection. While this technically is adequate for the evaluation of sensitivity, it does not allow to evaluate the false positive rate of IB and CNB (i.e. how many tumors were diagnosed as sarcoma after CNB and IB, only for the diagnosis to be revised in a benign lesion after resection. The authors should (preferrably) include patients with a sarcoma diagnosis after CNB and IB but a benign diagnosis after resection in their analyses and discussion. If they are unable to do this, they should provide explain in detail why they could not perform these analyses in their Methods section, and also add this point in their limitations section.
- Methods: In order for the readers to be able to reproduce the authors’ results, they need to know and understand the indications that the authors used for CNB and IB. The authors are currently too vague (“the feasibility of CNB was assessed first”, “in cases where some differential diagnoses were established beforehand and where more material was deemed necessary”) – please provide more information on when you chose to perform a CNB and when an IB.
- Methods: the authors state that “2-3 tissue cylinders were sent for histopathological examination.” One of the main advantages of IB in the guidelines is that it allows for more tissue to be acquired, so that some of this tissue may be stored in tumor banks “enabling later analyses for research, depending on local regulations” (ESMO Guidelines). Did the authors acquire fresh frozen tissue samples for tumor banking with IB, CNB, or both?
- Methods, lines 111-112: this is a result and should be presented in the Results section of the manuscript.
- Results: the authors write that: "63 (13.3%) underwent biopsies because of recurrent sarcomas." I cannot understand why the authors would include patients with suspected recurrent sarcomas. While biopsy of such lesions is indeed often necessary, the pathologist have the advantage of being able to compare the tissue to the primary tumor - if they are alike, the dignity and the entity are the same, with (rare) differences in grading. For that reason, the biopsies of recurrent tumors have to be removed from the analyses of the authors, in order to avoid bias.
- Results: the authors provide no details as to whether the patients in their cohort underwent preoperative radiotherapy or chemotherapy. In soft tissue sarcomas, grading cannot be reliably determined in the surgical specimen after preoperative radio- or chemotherapy. If the authors had such cases, they should exclude them at the very least from the analysis of the grading.
- Results, lines 131-134: why did these patients undergo a wide resection?? In their methods section, the authors describe an algorithm explaining what they did with suspected lesions following a negative biopsy, a primary wide resection despite a benign diagnosis and inconspicuous radiology was not part of that algorithm. How do the authors rule out then, that other patients who also had a benign diagnosis and incospicuous radiological findings but were perhaps treated elsewhere did not have a sarcoma as well? This rather speaks against the authors argument to only include sarcomas and rather for them examining all biopsies that they performed, as most authors apparently choose to do.
- Results, lines 137-146: how exactly did the authors calculate the rates of 83.7% for CNB and 86.5% for IB? According to Figure 2, the results for IB should be much higher.
- Results: this is a retrospective study, not a prospective randomized study following a power analysis. The authors should not write n.s., but provide both the absolute p-values and the ratios of patients for each analysis (XX/XXX vs. YY/YYY).
- Results, lines 150-152: this is another generalization. Which analyses did the authors perform in detail? Did they have adequate numbers for all these analyses?
- Results, lines 162-164: the grading of most bone sarcomas (ewing sarcoma, conventional high-grade osteosarcoma) is part of the entity definition, which the authors should reflect.
- Discussion: again, the authors should please provide appropriate references after each statement not based on their results, and more details on the studies they discuss (see previous comments).
- Discussion, lines 190-191: did the authors always perform CNB under ultrasound guidance? If so, why do they say this improves the accuracy of CNB, despite the fact that they had a much lower accuracy with their CNBs?
- Discussion: one of the main arguments against CNB is that repeat biopsies lead to treatment delays, which may affect oncological outcome. Indeed, the authors themselves write in their introduction that "However, an accurate and timely diagnosis is essential for the timely start of the appropriate therapy." However, this aspect is not discussed at all by the authors, I would strongly recommend to rectify this.
- Discussion, lines 218-220: the authors should either present the results of these subgroup analyses showing the numbers of patients they could include, or remove this statement.
- Discussion: The authors state that: "As the higher risk of complications in IB vs CNB is well known [26], CNB appears favourable in comparison. The risk of local recurrence in dependence of the type of biopsy is also of significant importance." If that were the case, why did the authors not examine complications and local recurrence in their cohort? As previously stated, they should either support these statements with their own results, or avoid basing their conclusions on them (they have submitted an original research paper, not a review).
- Discussion, limitations: rather than just reading a list of the limitations of the manuscript, the readers are generally more interested in understanding why these limitations were unavoidable, and why they should place value on the authors' findings despite these limitations.
- Discussion: the authors should try to avoid repeating themselves and work on the structure of their discussion, taking the above comments into consideration.
Author Response
- Further issues that need to be addressed (both in the abstract and in the main manuscript) are: Introduction: the authors should provide appropriate references for their statements (e.g. lines 34-37, 40-44, 49-50 etc.).
Response: The reviewer is right. We added the following references in Background and Discussion, to support our statements: 2; 3; 6; 13; 17-18; 20-22.
- Introduction (and discussion): When the authors present the results of previous stud-ies, they tend to be very vague (e.g. "Stoker et al showed a very high sensitivity with core-needle biopsies in the diagnosis of musculoskeletal lesions"). I would recommend that the authors go through their introduction and discussion and provide more details when presenting previous studies (e.g. what is "very high" sensitivity and in what set-ting with how many patients did Stoker et al show this?).
Response: This will in fact enhance the readability of the manuscript. More detailed information is now provided:
Line 57 addition of “with 97% of primary correct diagnosis”
Line 276 addition of “(76.4% vs. 73%)“
Line 288 addition of “(32% vs. 0.8%)“
- Introduction: The authors mention that: "The Department of Orthopaedic Oncology is the longstanding constituent of a dedicated Sarcoma Center. The interdisciplinary approach for the discussion of all newly diagnosed sarcoma cases as well as of all cases with suspicious lesions but without positive histological findings on biopsy has proven very valuable over the years.". This statements are unnecessary and not supported by facts or literature - they could therefore be interpreted by some readers as advertisement. Please remove them.
Response: This statement (lines 65-69) was deleted.
- Introduction, lines 61-66 - definitions of terms should be included in the Methods section, not the introduction section of the manuscript. Furthermore - the authors mention benign and malignant tumors. In terms of dignity, how did they classify locally advanced/rarely metastasizing bone and soft tissue tumors, which are considered neither benign nor malignant?
Response: Thank you for your suggestion. The definitions were transferred from lines 61-66 in the capture Method in lines 118-123. The locally advanced/rarely metastasizing bone and soft tissue tumors had not been included in this study.
- Methods, lines 68-71: the authors appear to have only included patients who had the "definitive diagnosis of sarcoma" after resection. While this technically is adequate for the evaluation of sensitivity, it does not allow to evaluate the false positive rate of IB and CNB (i.e. how many tumors were diagnosed as sarcoma after CNB and IB, only for the diagnosis to be revised in a benign lesion after resection. The authors should (pre-ferrably) include patients with a sarcoma diagnosis after CNB and IB but a benign diagnosis after resection in their analyses and discussion. If they are unable to do this, they should provide explain in detail why they could not perform these analyses in their Methods section, and also add this point in their limitations section.
Response: The critic of the reviewer is right, we have to clarify that. During the observational period there had been no patients with a malignant biopsy and a final diagnosis of a non-malignant tumor in the resection specimen.
We added the following sentence in the results section (lines 180-181): “During the observational period there had been no patients with a malignant biopsy and a final diagnosis of a non-malignant tumor in the resection specimen.”
- Methods: In order for the readers to be able to reproduce the authors’ results, they need to know and understand the indications that the authors used for CNB and IB. The authors are currently too vague (“the feasibility of CNB was assessed first”, “in cases where some differential diagnoses were established beforehand and where more material was deemed necessary”) – please provide more information on when you chose to perform a CNB and when an IB.
Response: In generally, our preferred method was always the CNB. The indications for CT- or US-guiding, as well as for IB are described in lines 99-104. To make this more clear, we added the word “generally” in line 100.
- Methods: the authors state that “2-3 tissue cylinders were sent for histopathological examination.” One of the main advantages of IB in the guidelines is that it allows for more tissue to be acquired, so that some of this tissue may be stored in tumor banks “enabling later analyses for research, depending on local regulations” (ESMO Guidelines). Did the authors acquire fresh frozen tissue samples for tumor banking with IB, CNB, or both?
Response: This is a very important issue. In general, we did not send any tissue of a CNB to a tissue bank. We are afraid that in this case, the tissue sent to pathology might not be sufficient for a final diagnosis and treatment of the patient might be delayed. Because of the alteration of the tissue in the resected specimen after neoadjuvant therapy, also this tissue is not appropriate for a tissue bank. We included the following statement in the discussion section (lines 236-238):
“The authors are well aware of the fact that due to the small sample sizes obtained with CNB a major disadvantage of this strategy is a loss of vital tissue for research i.e. storage in a tissue bank.”
- Methods, lines 111-112: this is a result and should be presented in the Results section of the manuscript.
Response: This sentence was transferred to line 155 in the results section.
- Results: the authors write that: "63 (13.3%) underwent biopsies because of recurrent sarcomas." I cannot understand why the authors would include patients with suspected recurrent sarcomas. While biopsy of such lesions is indeed often necessary, the pathologist have the advantage of being able to compare the tissue to the primary tumor - if they are alike, the dignity and the entity are the same, with (rare) differences in grading. For that reason, the biopsies of recurrent tumors have to be removed from the analyses of the authors, in order to avoid bias.
Response: Thank you for your suggestion. In our opinion, the biopsy in cases of suspected sarcoma recurrence is in many cases challenging. Obtaining a relevant tissue sample is sometimes more difficult compared to primary tumors. In many cases, it is a challenge, to identify neoplastic cells in scar tissue (sometimes after irradiation).
However, the reviewer is absolutely right in one important issue. The determination of the entity in the recurrent tumor is easier, if the tissue sample of the primary tumor is available. Also in this case we observed changes from a more defined lesion to a undifferentiated sarcoma. Therefore, we think, the inclusion of recurrent sarcomas causes no bias.
- Results: the authors provide no details as to whether the patients in their cohort underwent preoperative radiotherapy or chemotherapy. In soft tissue sarcomas, grading cannot be reliably determined in the surgical specimen after preoperative radio- or chemotherapy. If the authors had such cases, they should exclude them at the very least from the analysis of the grading.
Response: Thank you for this issue, which is important. We made a data correction and new evaluation. In lines 123-126 following sentences were added: “The grading of sarcoma was not allowed for every sarcoma subtype according WHO classification. The classification of grading in neoadjuvant treated sarcomas was not possible. These cases were excluded of sensitivity evaluation of grading.”
Lines 185-186 the sentence was added: “187 of 472 biopsies (39.6%) were excluded of the grading evaluation because of neoadjuvant therapy.” In Line 186 we added “… , as well as possible in depending of entity”. The results in line 187-188 were corrected. The last row in new table 2 was removed. In lines 213-215 the following sentence was added: “The evaluation of grading was done after exclusion of osteosarcomas and Ewing sarcomas because of neoadjuvant therapies and was hence limited to chondrosarcomas with significantly higher sensitivity for IB (p=0.024).”
- Results, lines 131-134: why did these patients undergo a wide resection?? In their methods section, the authors describe an algorithm explaining what they did with suspected lesions following a negative biopsy, a primary wide resection despite a benign diagnosis and inconspicuous radiology was not part of that algorithm. How do the authors rule out then, that other patients who also had a benign diagnosis and incospicuous radiological findings but were perhaps treated elsewhere did not have a sarcoma as well? This rather speaks against the authors argument to only include sarcomas and rather for them examining all biopsies that they performed, as most authors apparently choose to do.
Response: The reviewer is right. This paragraph (lines 158-162) was rewritten to:
“In 13 cases (3.1%), there were no indications of a malignant tumor in the histopathological examination of the tissue sample. In an interdisciplinary discussion based on clinical, radiological and pathological findings a malignant diagnosis was suspected. These cases underwent primary wide resection with the final diagnosis of a sarcoma.”
to make that more clear.
- Results, lines 137-146: how exactly did the authors calculate the rates of 83.7% for CNB and 86.5% for IB? According to Figure 2, the results for IB should be much higher.
Response: The reviewer is again right; this difference was not clearly explained. A small part of biopsies had a benign histology and the cases were resected followed by the histology of a sarcoma (false-negative biopsies as described above). The explanation was included between lines 156-160.
In line 160-161 we added the sentence: “In an interdisciplinary discussion based on clinical, radiological and pathological findings, a malignant diagnosis was suspected.”
Figure 2 was replaced by a corrected one.
- Results: this is a retrospective study, not a prospective randomized study following a power analysis. The authors should not write n.s., but provide both the absolute p-values and the ratios of patients for each analysis (XX/XXX vs. YY/YYY).
Response: The reviewer is right. We provided this in positions: lines 169; 183-184; 187-188; 192; 199-200; 220; Table 2.
- Results, lines 150-152: this is another generalization. Which analyses did the authors perform in detail? Did they have adequate numbers for all these analyses?
Response: The reviewer is again right.
The performed in each entity a comparison in respect to dignity, entity and grading. In respect to grading please read the next paragraph. To clarify this in more details we included a new Table 1. Regarding the power we now provided the number of cases in the table for giving the reader the chance to decide this issue for his own. We only find a significant difference in chondrosarcoma. Whether this is caused by a lack of power in all other entities is not be determined.
- Results, lines 162-164: the grading of most bone sarcomas (ewing sarcoma, conventional high-grade osteosarcoma) is part of the entity definition, which the authors should reflect.
Response: Thank you for your suggestion. We added followed information for better understanding: line 186 “, as well as possible in depending of entity” and line 189-190 “(if feasible for the entity)”. This should reflect the issue that not every sarcoma type has a grading system.
- Discussion: again, the authors should please provide appropriate references after each statement not based on their results, and more details on the studies they discuss (see previous comments).
Response: Done as proposed, see above.
- Discussion, lines 190-191: did the authors always perform CNB under ultrasound guidance? If so, why do they say this improves the accuracy of CNB, despite the fact that they had a much lower accuracy with their CNBs?
Response: Thank you for this important comment. In the original version, we wrote: “In addition, and in order to improve the accuracy of CNB, we perform these biopsies under ultrasound guidance.” This is not correct. In this paragraph we described the procedures and problems other authors described in using CNB. In selected difficult cases in respect to tumor size or location, we use US guidance. This is absolutly different from what was written before. Thank you once again! Therefore, we changed the whole sentence to (line 244-246):
“Our standard instrument for CNB is a 14-gauge Tru-Cut needle, according to the recommended guidelines [2]. In addition, and in order to improve the accuracy of CNB, we perform nowadays more frequently in selected cases (small tumor, non-palpable location) these biopsies under ultrasound guidance.”
- Discussion: one of the main arguments against CNB is that repeat biopsies lead to treatment delays, which may affect oncological outcome. Indeed, the authors themselves write in their introduction that "However, an accurate and timely diagnosis is essential for the timely start of the appropriate therapy." However, this aspect is not discussed at all by the authors, I would strongly recommend to rectify this.
Response: Thank you for your suggestion. We added the paragraph between lines 247-252: “The argument of faster diagnostic procedure by CNB relativized at least in 14% of patients. The repetition of the new biopsy is normally within one week possible. On another hand, there is no evidence for the influence of symptom duration on the oncologic outcome: no of citied studies was able to show the negative effect of longer symptom duration on the overall survival of sarcoma patients [20-22].”
- Discussion, lines 218-220: the authors should either present the results of these subgroup analyses showing the numbers of patients they could include, or remove this statement.
Response: See above, now included Table 1.
- Discussion: The authors state that: "As the higher risk of complications in IB vs CNB is well known [26], CNB appears favourable in comparison. The risk of local recurrence in dependence of the type of biopsy is also of significant importance." If that were the case, why did the authors not examine complications and local recurrence in their cohort? As previously stated, they should either support these statements with their own results, or avoid basing their conclusions on them (they have submitted an original research paper, not a review).
Response: The paragraph (lines 282-291) reads now:
In summary, we did not observe a significant difference between the 2 types of biopsies in this study. As the higher risk of complications in IB vs CNB is well known [35], CNB appears favourable in comparison. The risk of local recurrence in dependence of the type of biopsy is also of significant importance. Barrientos-Ruiz at al. analysed the oncological outcomes of 180 sarcoma patients with different kinds of biopsies and the contaminations of biopsy tracts. Their results were obvious: The contamination of the biopsy tracts was significantly higher in the cases of IB (32% vs. 0.8%) [36] and caused a higher number of local recurrences [37]. The lower risk of local recurrence after CNB was confirmed in other studies [38,39]. At most musculoskeletal oncology centers, CNB is therefore favoured over IB [12,13,36].
“(32% vs. 0.8%)” was introduced in line 288.
This study did not focus on the issues local recurrence or complication. This is right. However, comparing both types of biopsies one can not ignore those issues. For that, we included the citations and the numbers to give the reader the chance to make his own opinion on this.
- Discussion, limitations: rather than just reading a list of the limitations of the manuscript, the readers are generally more interested in understanding why these limitations were unavoidable, and why they should place value on the authors' findings despite these limitations.
Response: Absolutely right. We changed as follows:
Lines 294-296: “The primary biopsy method is CNB, when appropriate guided by sonography or computed tomography. Because of this, the patient group included in our study is subject to a certain degree of selection bias, which we acknowledge as a limitation.”
Was changed to: “As stated in the method section due to the preselection of cases with difficult differential diagnosis to IB a certain degree of selection bias has to be acknowledged as a limitation.” (lines 296-298)
Lines 300-301: the sentence “In addition, the retrospective design of our study limits the generalizability of our results.” was changed to:
“In addition the retrospective non-randomized design of this study limits the power of our findings.” (Lines 301-302)
Line 298-299: “Since benign lesions were not included, false positive results could not be evaluated.” This is wrong. We did not have any false positive biopsies. This sentence was hence deleted.
Lines 308-309 a sentence was added: “Comparing the rate of local recurrence and complications of both types of biopsies are necessary in future studies.”
- Discussion: the authors should try to avoid repeating themselves and work on the structure of their discussion, considering the above comments.
Response: We apologize for that and hope, that the revised manuscript is improved also in this respect.
Reviewer 2 Report
The manuscript consists of a retrospective comprehensive comparative study of the incisional or core-needle biopsy as a diagnostic method in a large number of 417 patients with soft tissue and bone sarcomas from a specialized center with high volume of patients. The analysis incudes both primary tumors and recurrenies. Information of the diagnostic spectrum and on the success and failure rate and follow up is provided.
The series presented is large, robust and provide a good comparison between incisional and core-needle biopsies in the diagnosis of soft tissue and bone tumors. The results are in favor of the application of the core-needle biopsy considering lower complications risk and better availability of this sampling method.
One minor comment:
In Results line 170: “So in total, about 15% of all primary biopsies returned false negative results (i.e.: benign or no tumor tissue)…”.
The non-diagnostic category - “no tumor tissue” use to be labeled as “insufficient”. Could you please clarify in how many cases of those of false negative category the sampling tissues were insufficient?
Author Response
- In Results line 170: “So in total, about 15% of all primary biopsies returned false negative results (i.e.: benign or no tumor tissue)…”. The non-diagnostic category - “no tumor tissue” use to be labeled as “insufficient”. Could you please clarify in how many cases of those of false negative category the sampling tissues were insufficient?
We added a sentence in Line 155: “In total, 51 patients needed 2, 2 patients 3 biopsies each.”
Round 2
Reviewer 1 Report
The authors have performed several revisions, unfortunately however, they have not addressed several important issues raised in the first review:
-Abstract: The authors still conclude about CNB being the diagnostic method of choice. As previously written, the authors did not perform a prospective, randomized study, but a retrospective analysis. As a result of this study design, they can actually only conclude that their algorithm for choosing to perform CNB and IB appears to be adequate for most aspects of diagnostic accuracy, although patients undergoing CNB had a significantly more repeat biopsies, something that the authors should emphasize more in their conclusions. What they cannot do is draw conclusions regarding the "diagnostic method of choice" in patients with bone and soft tissue sarcomas in general. The same applies to the the main manuscript, lines 284-294 - the authors need to rephrase.
- Methods: in their response to the reviewer, the authors wrote that: "the locally advanced/rarely metastasizing bone and soft tissue tumors had not been included in this study." They need to provide this information in the manuscript and explain why this was the case.
- The following comment is also inadequately addressed:
- Results: the authors write that: "63 (13.3%) underwent biopsies because of recurrent sarcomas." I cannot understand why the authors would include patients with suspected recurrent sarcomas. While biopsy of such lesions is indeed often necessary, the pathologist have the advantage of being able to compare the tissue to the primary tumor - if they are alike, the dignity and the entity are the same, with (rare) differences in grading. For that reason, the biopsies of recurrent tumors have to be removed from the analyses of the authors, in order to avoid bias.
The authors write: "Thank you for your suggestion. In our opinion, the biopsy in cases of suspected sarcoma recurrence is in many cases challenging. Obtaining a relevant tissue sample is sometimes more difficult compared to primary tumors. In many cases, it is a challenge, to identify neoplastic cells in scar tissue (sometimes after irradiation).
However, the reviewer is absolutely right in one important issue. The determination of the entity in the recurrent tumor is easier, if the tissue sample of the primary tumor is available. Also in this case we observed changes from a more defined lesion to a undifferentiated sarcoma. Therefore, we think, the inclusion of recurrent sarcomas causes no bias." The authors need to either present data from the literature supporting their opinion, or revise their cohort, as previously suggested.
- The authors write that: "We added the paragraph between lines 247-252: “The argument of faster diagnostic procedure by CNB relativized at least in 14% of patients. The repetition of the new biopsy is normally within one week possible. On another hand, there is no evidence for the influence of symptom duration on the oncologic outcome: no of citied studies was able to show the negative effect of longer symptom duration on the overall survival of sarcoma patients [20-22].” While that may be the case, there are studies demonstrating that repeat biopsies e.g. in osteosarcoma are associated with worse results for the patients (https://pubmed.ncbi.nlm.nih.gov/21030381/), something which the authors need to address in their discussion.
The authors reply that: "Their results were obvious: The contamination of the biopsy tracts was significantly higher in the cases of IB (32% vs. 0.8%) [36] and caused a higher number of local recurrences [37]. The lower risk of local recurrence after CNB was confirmed in other studies [38,39]. At most musculoskeletal oncology centers, CNB is therefore favoured over IB [12,13,36]." They should remove the word obvious (not a scientific approach), rephrase "caused a higher number..." to "were associated with a higher number..." (the cited study was retrospective and as such unable to identify causes, but only associations) and "At most musculoskeletal oncology centers..." to "At many musculoskeletal oncology centers..." as there are no data on which biopsy "most" musculoskeletal oncology centers perform.
Author Response
Thank you very much for the further attentive comments of the second reviewer.
-Abstract: The authors still conclude about CNB being the diagnostic method of choice. As previously written, the authors did not perform a prospective, randomized study, but a retrospective analysis. As a result of this study design, they can actually only conclude that their algorithm for choosing to perform CNB and IB appears to be adequate for most aspects of diagnostic accuracy, although patients undergoing CNB had a significantly more repeat biopsies, something that the authors should emphasize more in their conclusions. What they cannot do is draw conclusions regarding the "diagnostic method of choice" in patients with bone and soft tissue sarcomas in general. The same applies to the the main manuscript, lines 284-294 - the authors need to rephrase.
Thank you for the suggestion. We replaced in abstract line 37-38: “diagnostic method of choice” by “a valid and favourable method”.
Further: lines 297-298 “should be primary” was replaced by “is a valid and favourable method”; line 299 was added “(taking into account the higher repetition rate of CNB)”; lines 305-306 “should be method of choice” was replaced by “could be a more favourable method”.
- Methods: in their response to the reviewer, the authors wrote that: "the locally advanced/rarely metastasizing bone and soft tissue tumors had not been included in this study." They need to provide this information in the manuscript and explain why this was the case.
Lines 69-71 was rewritten. “Inclusion criteria were: -focusing on sarcomas the definitive diagnosis of a primary or locally recurrent soft tissue or bone sarcoma of the extremities, the pelvis and the trunk after resection at our center. All benign and intermediate lesions had been excluded.”
- The following comment is also inadequately addressed:
Results: the authors write that: "63 (13.3%) underwent biopsies because of recurrent sarcomas." I cannot understand why the authors would include patients with suspected recurrent sarcomas. While biopsy of such lesions is indeed often necessary, the pathologist have the advantage of being able to compare the tissue to the primary tumor - if they are alike, the dignity and the entity are the same, with (rare) differences in grading. For that reason, the biopsies of recurrent tumors have to be removed from the analyses of the authors, in order to avoid bias.
The authors write: "Thank you for your suggestion. In our opinion, the biopsy in cases of suspected sarcoma recurrence is in many cases challenging. Obtaining a relevant tissue sample is sometimes more difficult compared to primary tumors. In many cases, it is a challenge, to identify neoplastic cells in scar tissue (sometimes after irradiation).
However, the reviewer is absolutely right in one important issue. The determination of the entity in the recurrent tumor is easier, if the tissue sample of the primary tumor is available. Also in this case we observed changes from a more defined lesion to a undifferentiated sarcoma. Therefore, we think, the inclusion of recurrent sarcomas causes no bias." The authors need to either present data from the literature supporting their opinion, or revise their cohort, as previously suggested.
We added the paragraph between lines 264-270: “The inclusion of local recurrences in this study has to be discussed. Knowing the primary tumor might facilitate the final diagnosis. The diagnosis of the entity in recurrent tumor might be easier. However, in some cases we observed changes from a more distinct lesion to an undifferentiated sarcoma. Additional, the previous therapy may change the biology of the tumor, compared to primary sarcoma. The biopsy of suspected recurred tumor is recommended for these reasons [36].”
- The authors write that: "We added the paragraph between lines 247-252: “The argument of faster diagnostic procedure by CNB relativized at least in 14% of patients. The repetition of the new biopsy is normally within one week possible. On another hand, there is no evidence for the influence of symptom duration on the oncologic outcome: no of citied studies was able to show the negative effect of longer symptom duration on the overall survival of sarcoma patients [20-22].” While that may be the case, there are studies demonstrating that repeat biopsies e.g. in osteosarcoma are associated with worse results for the patients (https://pubmed.ncbi.nlm.nih.gov/21030381/), something which the authors need to address in their discussion.
The reviewer is right. We added between the lines 226-230: “Andreou et al. reported the inferior results (higher rate of local recurrences: 4.2% vs 10.1%; p=0.001) in patients, who underwent biopsies outside experienced centers [20]. The repetition of a biopsy could influence the outcome of treated patients negatively according to these results. However, this result is mainly based on IB with a higher risk of contamination.“
The authors reply that: "Their results were obvious: The contamination of the biopsy tracts was significantly higher in the cases of IB (32% vs. 0.8%) [36] and caused a higher number of local recurrences [37]. The lower risk of local recurrence after CNB was confirmed in other studies [38,39]. At most musculoskeletal oncology centers, CNB is therefore favoured over IB [12,13,36]." They should remove the word obvious (not a scientific approach), rephrase "caused a higher number..." to "were associated with a higher number..." (the cited study was retrospective and as such unable to identify causes, but only associations) and "At most musculoskeletal oncology centers..." to "At many musculoskeletal oncology centers..." as there are no data on which biopsy "most" musculoskeletal oncology centers perform.
We rephrased the sentences, as suggested.
Round 3
Reviewer 1 Report
I would still recommend that the authors explain their reasoning for excluding locally aggressive and rarely metastasizing tumors, as previously commented:
"- Methods: in their response to the reviewer, the authors wrote that: "the locally advanced/rarely metastasizing bone and soft tissue tumors had not been included in this study." They need to provide this information in the manuscript and explain why this was the case. Lines 69-71 was rewritten. “Inclusion criteria were: -focusing on sarcomas the definitive diagnosis of a primary or locally recurrent soft tissue or bone sarcoma of the extremities, the pelvis and the trunk after resection at our center. All benign and intermediate lesions had been excluded.” The authors still do not explain why they chose to do this.
- Editing of language and style is necessary prior to publication.
Author Response
I would still recommend that the authors explain their reasoning for excluding locally aggressive and rarely metastasizing tumors, as previously commented:
"- Methods: in their response to the reviewer, the authors wrote that: "the locally advanced/rarely metastasizing bone and soft tissue tumors had not been included in this study." They need to provide this information in the manuscript and explain why this was the case. Lines 69-71 was rewritten. “Inclusion criteria were: -focusing on sarcomas the definitive diagnosis of a primary or locally recurrent soft tissue or bone sarcoma of the extremities, the pelvis and the trunk after resection at our center. All benign and intermediate lesions had been excluded.” The authors still do not explain why they chose to do this.
As stated in the titel:
Relative Sensitivity of Core-Needle Biopsy and Incisional Biopsy in the Diagnosis of Musculoskeletal Sarcomas.
this study is intented to investigate the results of biopsies in soft tissue and bone sarcomas. We understand, that the reviewer would be much more satisfied if we could provide details also to results in locally aggressive intermediate tumors. But as it is, we did not collect and analyze this data, because those lesions had not been included in our patients cohort. So we have to admit that this investigation would be interesting, could provide a much more detailed view on biopsy in musculoskeletal tumors but we are very sorry that this study is unable to give any answers to this.
We take the point and will do a review of our data for those benign lesion in the coming year.
- Editing of language and style is necessary prior to publication.
The article has been reviewed and adjusted again.
Round 4
Reviewer 1 Report
The manuscript can now be published.
Author Response
Thank you for your time and comments.